# Characterization of COVID-19 Hospitalized Patients in Three United States Electronic Health Record Databases

**DOI:** 10.3390/pathogens12030390

**Published:** 2023-03-01

**Authors:** Patrick Saunders-Hastings, Cindy Ke Zhou, Shayan Hobbi, Eva Boyd, Patricia Lloyd, Nader Alawar, Timothy Burrell, Jeff Beers, Tainya C. Clarke, Aaron Z. Hettinger, Hui-Lee Wong, Azadeh Shoaibi

**Affiliations:** 1Gevity Consulting Inc., part of Accenture, Ottawa, ON K1P 1A4, Canada; 2Center for Biologics Evaluation and Research, US Food and Drug Administration, Silver Spring, MD 20993, USA; 3IBM Consulting, Bethesda, MD 20814, USA; 4Center for Biostatistics, Informatics and Data Science, MedStar Health Research Institute, Hyattsville, MD 20782, USA

**Keywords:** COVID-19, hospitalization, EHR, epidemiology, treatment, United States

## Abstract

COVID-19 infections have contributed to substantial increases in hospitalizations. This study describes demographics, baseline clinical characteristics and treatments, and clinical outcomes among U.S. patients admitted to hospitals with COVID-19 during the prevaccine phase of the pandemic. A total of 20,446 hospitalized patients with a positive COVID-19 nucleic acid amplification test were identified from three large electronic health record databases during 5 February–30 November 2020 (Academic Health System: *n* = 4504; Explorys; *n* = 7492; OneFlorida: *n* = 8450). Over 90% of patients were ≥30 years of age, with an even distribution between sexes. At least one comorbidity was recorded in 84.6–96.1% of patients; cardiovascular and respiratory conditions (28.8–50.3%) and diabetes (25.6–44.4%) were most common. Anticoagulants were the most frequently reported medications on or up to 28 days after admission (44.5–81.7%). Remdesivir was administered to 14.1–24.6% of patients and increased over time. Patients exhibited higher COVID-19 severity 14 days following admission than the 14 days prior to and on admission. The length of in-patient hospital stay ranged from a median of 4 to 6 days, and over 85% of patients were discharged alive. These results promote understanding of the clinical characteristics and hospital-resource utilization associated with hospitalized COVID-19 over time.

## 1. Introduction

The COVID-19 pandemic began in December 2019 and has extended into 2023, resulting in significant illness and mortality. As of 6 January 2023, roughly 101 million COVID-19 cases had been reported in the United States (U.S.) alone, and this number is sure to be an underestimate [1]. COVID-19 infections have also driven substantial increases in hospitalizations and deaths, with over 5.7 million COVID-19 admissions and over 1.09 million deaths reported between 1 August 2020 (date of consistent reporting), and 3 January 2023 [1].

Descriptive analyses of mild, moderate, and severe COVID-19 cases have been essential to enhancing the understanding of the clinical course of infection, informing clinical management and public health strategies to combat and contain the disease. For example, Lucar and colleagues [2] conducted an observational study of 100 patients hospitalized in Mississippi, reporting a median age of 59 years and common comorbidities including hypertension, obesity, and diabetes among a patient population with a median symptom duration of 7 days. A 2022 study [3], meanwhile, characterized 100 Canadian patients admitted during the first wave of the pandemic, noting the multisystemic and (often) persistent nature of severe COVID-19 infection, leading to respiratory, neurologic, cardiac, and thrombotic complications. However, past studies have often been limited to small or geographically constrained patient populations with little or no longitudinal follow-up. This study seeks to address this knowledge gap by assessing COVID-19 outcomes longitudinally among three large, distributed patient cohorts.

Pandemic preparedness and response are priorities for the Food and Drug Administration’s (FDA) Center for Biologics Evaluation and Research (CBER) Biologics Effectiveness and Safety (BEST) Initiative. As part of the BEST network, this study leveraged data from three electronic health record (EHR) databases, and described the demographics, baseline clinical characteristics and treatments, and clinical outcomes during hospitalization among U.S. patients admitted to the hospital with at least one positive SARS-CoV-2 nucleic acid amplification test (NAAT) in the prevaccine phase of March to November 2020.

## 2. Materials and Methods

### 2.1. Study Population

This was a retrospective cohort study of hospitalized COVID-19-positive patients, confirmed by SARS-CoV-2 NAAT within 14 days of hospital admission during 5 February–30 November 2020. The start of the study period was selected based on the approval date of the first FDA-authorized SARS-CoV-2 NAAT for COVID-19 diagnosis. The study end date represents the last full month of data available before the first COVID-19 vaccine was given FDA emergency use authorization on 11 December 2020.

The study population was drawn from three EHR databases across the U.S. (Academic Health System (AHS), Explorys, and OneFlorida), serving approximately 78 million patients per year. AHS provides a live EHR connection to a health system that serves patients in the Mid-Atlantic area of the U.S., providing EHR data on approximately 6 million patients from 2010 to the present, and about 1.5–2 million active patients annually, on average. The OneFlorida Clinical Research Consortium is an EHR consortium spanning the state of Florida, containing patient-level data from public and private health care systems. In total, the data trust contains EHR data for about 6 million Floridians or about 75% of Floridians from 2012 to the present, with about 2–3 million patients annually, on average. IBM Explorys is an aggregated EHR database of more than 39 health system partners’ EHR data, spanning academic centers and community practices. Explorys represents records of approximately 70 million patients annually, with most encounters in the past year concentrated in Ohio, Louisiana, Georgia, Florida, and New York.

### 2.2. Eligibility Criteria

All hospitalized patients (at least 1-day length of stay) who had a positive SARS-CoV-2 NAAT test and admission and discharge date or status to construct a hospitalization episode were included for analysis.

Hospital stays missing admission or discharge dates were excluded from analyses, as hospitalization episodes could not be constructed. This exclusion was only relevant to the Explorys database, as all AHS or OneFlorida admissions had admission and discharge dates.

### 2.3. Clinical Characteristics

The patient characteristics of interest included demographics at admission (age, sex, and recorded race/ethnicity), comorbidities, symptoms and clinical signs, concomitant medications, COVID-19 severity, and hospitalization measures (in-patient hospital length of stay, intensive care unit (ICU) admission and assisted ventilation use, dates of hospitalization, and hospital discharge disposition).

Comorbidities were assessed up to 365 days prior to hospital admission and separately on admission. Symptoms were assessed up to 14 days prior to admission to capture infections detected in an outpatient setting prior to admission, while patient COVID-19 symptoms were assessed 14 days prior to admission, on admission, and 14 days following admission. Comorbidities, symptoms, and clinical signs were identified from structured EHR data and grouped based on clinical categories mapped to the International Classification of Diseases, 10th Revision, Clinical Modification (ICD-10-CM) codes, which were then mapped to Systematized Nomenclature of Medicine Clinical Terms (SNOMED-CT) codes using the Observational Health Data Sciences and Informatics (OHDSI) Observational Medical Outcomes Partnership (OMOP) vocabulary mappings. Similarly, concomitant medications were assessed on and up to 28 days following hospital admission and categorized and mapped to National Drug Code (NDC), RxNorm, and Cerner Multum codes, with clinician review. The code lists used from these vocabularies can be referenced in Appendix A.

COVID-19 severity was evaluated 14 days prior to admission, on the day of admission, and 14 days following admission. Severity was categorized as mild, moderate, severe, or critical, based on guidelines released by FDA [4], with details provided in Appendix A. Severity categories were based on clinical characteristics and laboratory data available for hospitalized patients.

Hospitalization measures were assessed during the hospitalization episode. Hospital discharge disposition was stratified into five categories: discharged alive, transferred to another hospital, transferred to a long-term care (LTC) facility, and discharge status other/unknown. Patients discharged to another hospital or discharged to an LTC facility were included within the discharged alive category. In-hospital mortality was available in AHS and OneFlorida, while ICU and ventilation data were only available in AHS. ICU episodes were constructed based on bed-level, time-stamped patient location data, and were defined as the time from the first ICU encounter to the last ICU encounter. Ventilation status was defined as any form of assisted ventilation and was based on structured Cerner observation codes.

### 2.4. Statistical Analyses

Anonymized data used for analysis was extracted from electronic health record data previously collected during routine healthcare interactions and made available for research purposes. Analyses were conducted separately for each data source, and all variables were stratified by month for March–November 2020. In keeping with standard practice for descriptive statistics, frequencies and proportions were calculated for categorical variables, and the mean (with standard deviation) or median (with 5th–95th or 25th–75th percentile ranges) was calculated for continuous variables. Data were analyzed using Python (Version 3.9) and Microsoft Excel^®^ (2019).

## 3. Results

### 3.1. Demographics

No hospital admissions were identified until March 2020. A total of 20,446 hospitalized patients with at least one positive COVID-19 NAAT were identified across three databases (AHS: 4504, Explorys: 7492, OneFlorida: 8450) during the study period (Table 1). An additional 7206 inpatient admissions were identified in Explorys but were excluded due to the absence of admission and/or discharge dates. Over 90% of patients were 30 years or older, with those 50–64 years of age making up the largest proportion of the patient cohort (27.5–29.4% across databases). The median (25th–75th percentile) age at the time of admission was reported to be 62 (48–73), 64 (50–75), and 60 (45–73) years in AHS, Explorys, and OneFlorida, respectively. Sex was evenly distributed across databases. The most frequently recorded races were black (57.8%) in AHS and white in Explorys (62.3%) and OneFlorida (44.6%). More than 30% were recorded as Hispanic in OneFlorida, a larger proportion compared with the other two databases that did not exceed 10%. Patients most often self-reported their ethnicity as non-Hispanic (62.1–87.4% across databases), with the highest proportion reporting as Hispanic observed in OneFlorida (30.5%) and the highest proportion reporting “Unknown” ethnicity observed in AHS (12.2%). Additional information on patients’ characteristics is provided in Appendix A.

### 3.2. Comorbidities

Table 1 presents comorbidities up to 365 days prior to COVID-19-related hospitalization. Across the three databases, 84.6–96.1% of hospitalized patients had at least one diagnosis recorded. The most common comorbidities were cardiovascular and respiratory conditions (28.8–50.3% of patients across databases), followed by diabetes (25.6–44.4%). Smoking was common among patients in Explorys at 24.5%. Obesity was also common among hospitalized patients in both Explorys and OneFlorida, at 23.5% and 31.9%, respectively.

### 3.3. Medications

Concomitant medications reported on or up to 28 days after admission identified in each database are presented in Table 2, while monthly medication prescription data are included in Appendix A. Anticoagulants were the most frequently administered medication class across databases (44.5–81.7%). AHS administered anticoagulants (81.7%), azithromycin (37.4%), antiplatelets (28.0%), and dexamethasone (26.5%) most frequently during the 9-month study period. Explorys displayed a similar pattern, with anticoagulants (74.3%), dexamethasone (39.1%), antiplatelets (34.0%), and azithromycin (28.3%) being the most frequently administered. While the overall trend was similar in OneFlorida, the proportions of patients receiving each type of medication were lower than in the other databases: anticoagulants (44.5%), dexamethasone (23.9%), azithromycin (19.7%), and antiplatelets (19.6%).

Remdesivir was administered to 14.1–24.6% of patients and increased over time in all data sources (Table 2 and Appendix A). Remdesivir was not utilized in the beginning of the study period (March and April), but by November, it was being administered to 33.6%, 44.4%, and 39.7% of hospitalized COVID-19-positive patients in AHS, Explorys, and OneFlorida, respectively. Conversely, hydroxychloroquine and chloroquine orders peaked in March across the three databases (AHS: 79.1%, Explorys: 37.7%, and OneFlorida: 46.7%) before decreasing for the remainder of the study period. Anticoagulants were the most commonly prescribed medication throughout the study period across all databases (Appendix A).

### 3.4. Clinical Signs and Symptoms

The symptoms most frequently reported 14 days prior to or on admission among hospitalized COVID-19 patients in AHS, Explorys, and OneFlorida were dyspnea (36.9%, 29.7%, and 24.5%, respectively), fever (23.0%, 17.6%, and 17.2%, respectively), cough (18.0%, 21.3%, and 11.4%, respectively), and fatigue (10.1%, 19.7%, and 9.0%, respectively). Respiratory failure was the most reported condition in OneFlorida, present among 40.4% of patients (Table 3).

Data on the frequency of clinical signs and symptoms reported by month are presented in Appendix A. The relative frequency of observation of clinical signs and symptoms did not differ meaningfully throughout the study period within each data source, although there was a decrease in the frequency of all symptoms reported in AHS over time.

### 3.5. COVID-19 Severity

The severity of COVID-19 symptoms on or 14 days prior to admission and 14 days following admission is presented in Figure 1. In AHS, 42.9–51.7% of cases were classified as mild across the study period, and 27.5–33.2% were classified as critical 14 days prior to or on admission. Following admission, the proportion of mild cases decreased to 14.9–32.1%, while critical cases increased to 67.3–76.8% across the study period.

Hospitalizations identified in Explorys followed a similar pattern, with most cases prior to or on admission being classified as mild (69.6–81.7%) and decreasing following admission (11.2–22.9%). Critical cases were reported among 17.9–30.2% of patients prior to and on admission, increasing to 69.2–80.8% in the 14 days following admission.

The observed trends in severity differed in the OneFlorida data, with 32.4–48.7% of cases classified as mild and 51.3–67.6% of cases classified as critical prior to and on admission. The higher proportion of critical cases extended into the postadmission period (63.2–74.0%), while mild cases made up 27.1–36.8% of postadmission cases. Additional information on the proportion of mild, moderate, severe, and critical cases, by month and admission status, is provided in Figure 1.

### 3.6. Hospitalization Measures

Data on the distribution of hospitalization measures over time and by data source are presented in Figure 2. The median (5th–95th percentile) in-patient hospital length of stay was 6 days in AHS (1–32 days), 4 days in Explorys (1–22 days), and 5 days in OneFlorida (1–28 days). Length of stay peaked between March and April, before decreasing later in the study period.

ICU length of stay and ventilation data were available in the AHS database only, where 1,565 (34.7%) had an ICU stay at any time during their hospital stay. The median length of ICU stay was 5 days (5th percentile: <1 day (stay less than 24 h), 95th percentile: 29 days), decreasing from 8 days in March to 4 days in November. Mechanical ventilation was administered to 85.9% of the AHS study population during any point of hospitalization stay. The median number of days that a patient was on a ventilator was 6 (5th percentile: <1 day, 95th percentile: 32 days), with 75% of patients receiving ventilation for 11 days or less. Of patients admitted to the ICU, 91.8% were placed on a ventilator at some point during their ICU stay.

Over 85% of patients across databases were discharged alive with a length of stay ≤28 days (85.3% of patients in AHS, 94.3% in Explorys, and 91.2% in OneFlorida). In the OneFlorida database, 8.8% of patients were deceased within 28 days of their in-patient stay, compared with 14.7% in AHS (mortality data were not available for Explorys). Patient hospitalization, discharged alive, and in-hospital mortality were highest early in the study period.

## 4. Discussion

This study described the demographics, clinical characteristics, and hospital-resource utilization of 20,446 COVID-19-hospitalized patients identified between March and November 2020 in three large U.S. EHR databases (AHS: 4504; Explorys: 7492, OneFlorida: 8450). Patient demographics did not differ across databases, with a nearly even split between males and females and a median patient age ranging from 60 to 64 years. Previous studies of hospitalized patients early in the pandemic reported similar age range but noted that the majority of cases were male [5,6,7,8,9,10]. However, Roth and colleagues [11] observed an equal proportion of males and females hospitalized for COVID-19 in the U.S. Diabetes, cardiovascular (including hypertension), and respiratory conditions were the most frequently reported comorbidities across databases. This was especially notable in OneFlorida, where 50.3% of identified patients had cardiovascular and respiratory conditions reported in the 365 days prior to hospital admission. These findings align with comorbidities observed in past U.S. studies of patients with severe COVID-19, which have identified diabetes, hypertension, cardiovascular disease, asthma, and obesity as the most common comorbidities [12,13,14]. Further, studies have suggested that hospitalization rates are higher among those with multiple (≥3) comorbidities [15]. Fever, cough, and dyspnea were the frequently reported symptoms across all databases, consistent with clinical expectation and prior epidemiological findings [16].

Medications most frequently administered aligned with early COVID-19 treatment guidelines and recommendations [17], with anticoagulants (44.5–81.7% across databases), azithromycin (19.7–37.4%), dexamethasone (23.9–39.1%), and antiplatelets (19.6–34.0%) being commonly observed. However, azithromycin administered alone or in combination with chloroquine/hydroxychloroquine is not currently recommended based on the evidence available [17]. In comparison, Roth and colleagues [11] studied hospitalized COVID-19-positive patients identified from a cardiovascular disease registry, reporting that over 50% of patients were ordered azithromycin and a third of patients were nondexamethasone glucocorticoids. The observed differences could be due to the characteristics of the study patients. Temporal analyses of concomitant medications showed that remdesivir and dexamethasone use increased substantially during the study period, especially June and July, and hydroxychloroquine/chloroquine exhibited a marked decrease in March and April, which was sustained throughout the study period. This is consistent with an earlier medication study of COVID-19 patients hospitalized in California in March–December 2020 [18].

The trends of COVID-19 clinical severity remained stable across all databases over the study period but differed when stratified by 14 days prior to and on admission and 14 days following admission. Patients were most often characterized as having mild disease at admission, which could have been due to a lack of recorded information at admission necessary to classify severity. Across databases, 23.0–62.1% of patients hospitalized for COVID-19 were categorized as critically ill 14 days prior to and on admission. The proportion of patients classified as critically ill increased following admission across databases (to 69.6–74.6%), suggesting that disease progression continued after hospital admission, consistent with previous studies [19].

The length of in-patient hospital stay ranged from a median of 4 to 6 days across databases, comparable to previous studies [20]. Another study of U.S. patients hospitalized for COVID-19 reported a similar increase in in-hospital mortality between April and May 2020, although authors reported a higher in-hospital mortality rate (19.1% vs. 5% in the present study) [11]. ICU admissions (available in one data source) were also highest in April and May 2020. Roth and colleagues [11] observed a similar trend, hypothesizing that the rapid implementation of new isolation measures, personal protective measures, and therapeutics (nondexamethasone glucocorticoids and remdesivir) may explain the decrease in ICU admissions observed in June 2020 and beyond. While the present study identified no change in the administration of nondexamethasone glucocorticoids, remdesivir was not widely administered to patients admitted in March and April (only four patients received remdesivir across the three databases), which may have contributed to a higher rate of ICU admission.

## 5. Conclusions

Few studies have longitudinally assessed hospitalized COVID-19 patients in multiple EHR databases. The strengths of this study position the findings to examine a larger patient cohort identified and assessed over a longer period and across multiple databases, increasing the geographic representativeness of the patient cohort and improving the external validity and generalizability of findings. This study adds valuable insight into the clinical course of infection, patient characteristics and comorbidities, and hospital-resource utilization associated with severe COVID-19 infection over time. This study demonstrates the capability of the FDA BEST Initiative in conducting active surveillance and promoting public health.

This study includes several limitations. First, results of SARS-CoV-2 tests conducted outside of facilities covered by the three EHR systems may not have been captured in the study data. Therefore, the study may have underestimated the prevalence of hospitalized COVID-19-positive patients. Similarly, COVID-19-related hospitalizations in Explorys were likely underestimated, as 49% of inpatient admissions were excluded from the analysis due to missing discharge dates, which precluded construction of hospitalization episodes. Second, COVID-19 severity may have been similarly underestimated due to a lack of data necessary to classify patients as moderate, severe, or critical. Third, ICU admission data were only available in AHS, while in-hospital mortality data were only available in AHS and OneFlorida. Fourth, the reliability and validity of some patient characteristics may have been limited by the using ICD-10-CM and SNOMED-CT codes. Similarly, reliance on structured data elements for clinical features may have led to the under-reporting or misclassification of mild or unspecific COVID-19 symptoms, which could have impacted our assessment of symptoms. Recorded race/ethnicity may also not accurately reflect self-reported race/ethnicity. Fifth, patient cohorts were identified from three health databases representing populations from the Eastern and Mid-Atlantic United States and may not be representative of cases elsewhere in the country. Lastly, the presented findings are descriptive and, as such, do not take into account or adjust for potential confounding factors.

This study helps understand the clinical course of infection, characteristics of hospitalized COVID-19 patients, and hospital-resource utilization associated with severe COVID-19 diseases over time, with implications for how physicians monitor admitted patients for increases in severity during hospital stay and inform resource planning and triage efforts (such as for ventilators and medications). Further, recognizing that past studies [21,22] have found associations with ICU admission and feelings of stress, anxiety, isolation, and depression, both during hospital stay and after discharge, findings offer an opportunity to improve communication with hospitalized patients to reduce both physical and psychological health burden associated with COVID-19 hospitalization. Future studies may further evaluate predictors of population characteristics for severe COVID-19 outcomes, including via analytical statistics comparing patients admitted for COVID-19 infection with the general patient population.

## Figures and Tables

**Figure 1 pathogens-12-00390-f001:**
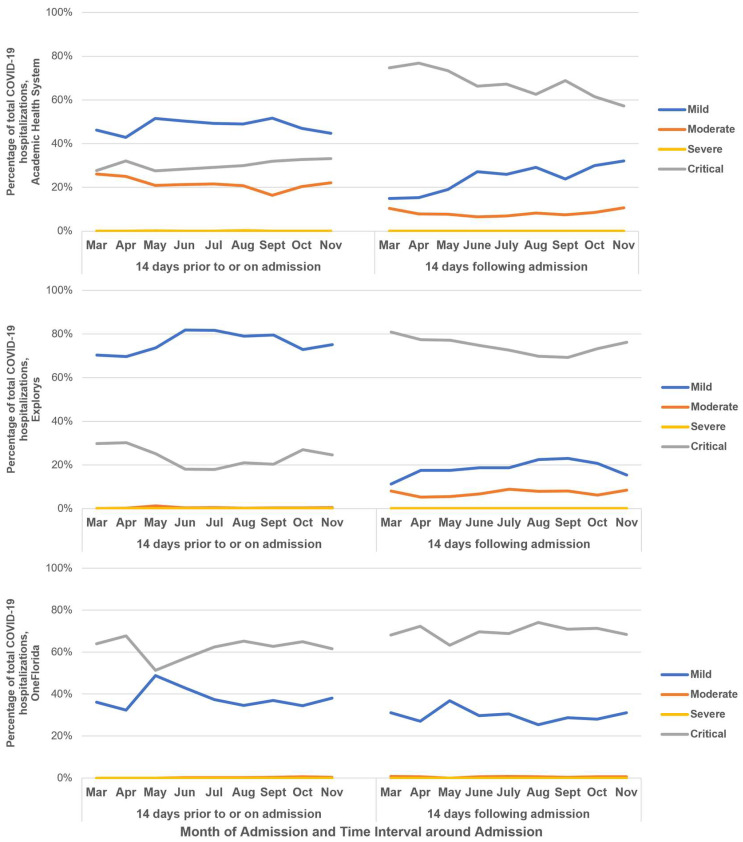
COVID-19 severity among hospitalized positive SARS-CoV-2 NAAT patients within 14 days of admission (March–November 2020).

**Figure 2 pathogens-12-00390-f002:**
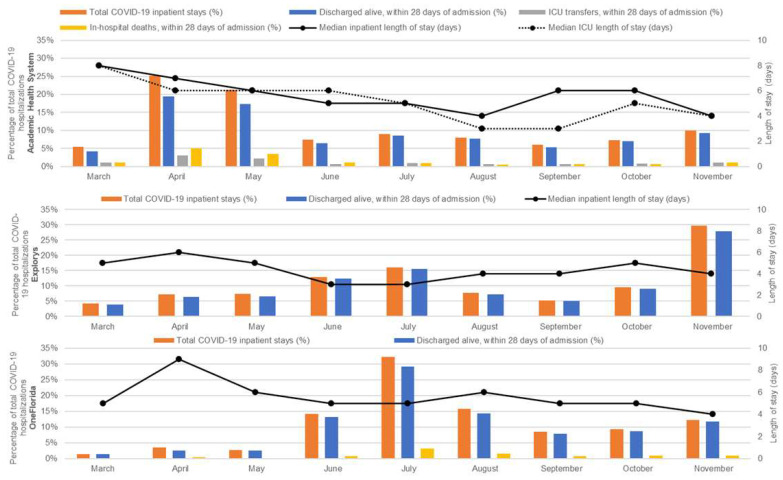
Hospitalization measures over time positive SARS-CoV-2 NAAT patients within 14 days of admission (March–November 2020).

**Table 1 pathogens-12-00390-t001:** Characteristics of SARS-CoV-2 NAAT-positive patients within 14 days of admission (March–November 2020).

Characteristics	Academic Health System (*N* = 4504)	Explorys (*N* = 7492)	OneFlorida (*N* = 8450)
**Age at Admission (Years)**			
Mean (Standard Deviation)	60.1 (18.1)	61.2 (18.9)	58.3 (19.6)
Median (Interquartile Range)	62 (4873)	64 (5075)	60 (4573)
Age at Admission (Age Group)	*n* (%) ^2^	*n* (%) ^2^	*n* (%) ^2^
0–17	10 (0.2)	113 (1.5)	153 (1.8)
18–29	290 (6.4)	460 (6.1)	600 (7.1)
30–49	963 (21.4)	1224 (16.3)	1888 (22.3)
50–64	1323 (29.4)	2060 (27.5)	2360 (27.9)
65–74	917 (20.4)	1635 (21.8)	1563 (18.5)
75–84	660 (14.7)	1243 (16.6)	1189 (14.1)
85+	341 (7.6)	750 (10)	697 (8.2)
Not Specified/Unknown	0 (0)	7 (0.1)	0 (0)
**Sex**			
Female	2229 (49.5)	3751 (50.1)	4217 (49.9)
Male	2275 (50.5)	3741 (49.9)	4233 (50.1)
**Recorded Race**			
American Indian or Alaska Native	10 (0.2)	0 (0)	10 (0.1)
Asian/Pacific Islander	58 (1.3)	101 (1.3)	141 (1.7)
Black/African American	2604 (57.8)	2272 (30.3)	2418 (28.6)
White/Caucasian	814 (18.1)	4664 (62.3)	3768 (44.6)
Other	915 (20.3)	444 (5.9)	1959 (23.2)
Unknown	103 (2.3)	92 (1.2)	154 (1.8)
**Recorded Ethnicity**			
Hispanic	383 (8.5)	691 (9.2)	2576 (30.5)
Non-Hispanic	3570 (79.3)	6550 (87.4)	5251 (62.1)
Unknown	551 (12.2)	272 (3.6)	623 (7.4)
**Comorbidities ^1^**			
Any diagnosis or problem list entry	4086 (90.7)	6339 (84.6)	8124 (96.1)
Cancer	293 (6.5)	1231 (16.4)	845 (10.0)
Cardiovascular and respiratory conditions	1296 (28.8)	3542 (47.3)	4253 (50.3)
Diabetes	1155 (25.6)	2575 (34.4)	3750 (44.4)
Immunosuppression	131 (2.9)	666 (8.9)	213 (2.5)
Kidney disease	615 (13.7)	2636 (35.2)	1592 (18.8)
Liver disease	140 (3.1)	422 (5.6)	391 (4.6)
Autoimmune disease	194 (4.3)	1185 (15.8)	608 (7.2)
Obesity	675 (15.0)	1764 (23.5)	2696 (31.9)
Smoking	138 (3.1)	1833 (24.5)	402 (4.8)

^1^ Comorbidities may equal >100% by data source due to patients having multiple comorbidities. ^2^ Proportions are out of all hospitalized patients with a positive NAAT within 14 days of admission in each data source (N).

**Table 2 pathogens-12-00390-t002:** Concomitant medications observed on or up to 28 days after admission (March–November 2020).

Medication Category ^1^	Academic Health System (*N* = 4504)	Explorys (*N* = 7492)	OneFlorida (*N* = 8450)
	*n* (%) ^2^	*n* (%) ^2^	*n* (%) ^2^
Angiotensin-converting enzyme (ACE) inhibitors	481 (10.7)	1202 (16.0)	836 (9.9)
Angiotensin II receptor blockers	126 (2.8)	225 (3.0)	109 (1.3)
Anticoagulant	3679 (81.7)	5570 (74.3)	3757 (44.5)
Antiplatelet	1262 (28.0)	2544 (34.0)	1660 (19.6)
Azithromycin	1684 (37.4)	2120 (28.3)	1663 (19.7)
Dexamethasone	1192 (26.5)	2931 (39.1)	2019 (23.9)
Nondexamethasone glucocorticoids	686 (15.2)	1439 (19.2)	896 (10.6)
HIV protease inhibitor (lopinavir, ritonavir)	4 (0.1)	9 (0.1)	2 (0.0)
Other HIV protease inhibitor	9 (0.2)	16 (0.2)	15 (0.2)
Hydroxychloroquine/chloroquine	747 (16.6)	403 (5.4)	232 (2.7)
IL-1 receptor antagonist	0 (0.0)	0 (0.0)	6 (0.1)
IL-6 receptor antagonist	178 (4.0)	148 (2.0)	449 (5.3)
Influenza antiviral	14 (0.3)	52 (0.7)	16 (0.2)
Remdesivir	633 (14.1)	1842 (24.6)	1371 (16.2)

^1^ Concomitant medications may sum to >100% by data source due to patients receiving multiple concomitant medications. ^2^ Proportions are out of all hospitalized patients with a positive NAAT within 14 days of admission in each data source (*N*).

**Table 3 pathogens-12-00390-t003:** Patient signs and symptoms (March–November 2020).

Signs and Symptoms Present up to 14 Days Prior to or on Admission ^1,2^	Academic Health System (*N* = 4504)	Explorys (*N* = 7492)	OneFlorida (*N* = 8450)
	*n* (%) ^3^	*n* (%) ^3^	*n* (%) ^3^
Chest discomfort/pain	248 (5.5)	1044 (13.9)	887 (10.5)
Chills	48 (1.1)	110 (1.5)	37 (0.4)
Cough	810 (18.0)	1595 (21.3)	966 (11.4)
Decreased appetite (anorexia)	65 (1.4)	175 (2.3)	120 (1.4)
Diarrhea	165 (3.7)	725 (9.7)	766 (9.1)
Dizziness/light-headed	113 (2.5)	344 (4.6)	146 (1.7)
Fatigue	454 (10.1)	1478 (19.7)	764 (9.0)
Fever	1036 (23.0)	1322 (17.6)	1450 (17.2)
Headache	86 (1.9)	272 (3.6)	252 (3.0)
Loss of smell/taste (anosmia/dysgeusia)	28 (0.6)	70 (0.9)	58 (0.7)
Multiorgan failure	45 (1.0)	551 (7.4)	1160 (13.7)
Myalgia	34 (0.8)	146 (1.9)	315 (3.7)
Myocarditis	0 (0.0)	6 (0.1)	26 (0.3)
Nausea or vomiting	281 (6.2)	812 (10.8)	494 (5.8)
Palpitations	12 (0.3)	80 (1.1)	68 (0.8)
Respiratory failure	219 (4.9)	1344 (17.9)	3413 (40.4)
Shortness of breath (dyspnea)	1663 (36.9)	2228 (29.7)	2074 (24.5)
Sore throat	36 (0.8)	152 (2.0)	84 (1.0)
Stomach/abdominal pain	252 (5.6)	640 (8.5)	625 (7.4)

^1^ Signs and symptoms may equal >100% by data source due to patients having multiple symptoms. Proportions are out of all hospitalized patients with a positive NAAT within 14 days of admission in each data source (*N*). ^2^ The same list of symptoms was queried for each database. The symptoms in this table are sorted alphabetically. ^3^ Proportions are out of all hospitalized patients with a positive NAAT within 14 days of admission in each data source (*N*).

## Data Availability

All data generated or analyzed during this study are included in this published article and its Appendix A.

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
