# Peer review of "Characterization of COVID-19 Hospitalized Patients in Three United States Electronic Health Record Databases"

_pathogens, 2023, doi:10.3390/pathogens12030390_

Round 1

Reviewer 1 Report

Following are the changes for the acceptance of the article for publication

1. The abstract should be revised into the sections of (i) background of the study (ii) methodology (iii) results (iv) conclusion

2.  the title of article is quite lengthy. It should be like Characterization of COVID-19 hospitalized patients in three United States electronic health record databases. the time range should be mentioned in abstract.

3.  Keywords are short. There must be few  more keywords

4.  Introduction is quite short. It should be extended with latest references. the background of the study should be enough to jusitfy the study.

5. Which sampling methodology was used. The statistical techniques used should be explained in one line and why did you use that technique particularly.

6. figures are not adjusted .for example it is difficult to see the figure 1. Moreover there are two figures with lable figure 1. Kindly change the figure numbers and adjust it in the explanation.

7.Conclusion section should also be extended with future perspectives and limitations of the study.

8. Grammatical and sentence structure of overall study should be checked for the mistakes

overall study is acceptable for publication but after considering the above mentioned changes.

Author Response

Reviewer #1

Following are the changes for the acceptance of the article for publication

  • Authors: We would like to thank the reviewer for taking the time to review and provide feedback on the manuscript. We greatly appreciate their comments and believe that the manuscript has improved as a result of their contribution. We have outlined below the actions taken in response to each individual comment; these are numbered as “R1.x” (for Reviewer 1 and the comment number).

R1.1. The abstract should be revised into the sections of (i) background of the study (ii) methodology (iii) results (iv) conclusion

  • Authors: We have revised the abstract structure as recommended.

R1.2. The title of article is quite lengthy. It should be like Characterization of COVID-19 hospitalized patients in three United States electronic health record databases. the time range should be mentioned in abstract.

  • Authors: We have revised the article title as recommended, keeping the time range in the Abstract (Methodology sub-section)

R1.3.  Keywords are short. There must be few more keywords

  • Authors: We have added additional keywords for a total of six (within the range of 3-10 requested by the journal).

R1.4.  Introduction is quite short. It should be extended with latest references. the background of the study should be enough to justify the study.

  • Authors: We have extended the introduction with recent references to clarify the justification for this study (i.e., to address geographical/sample size/follow-up limitations of past studies). While we draw on references from 2021 and 2022, it should be noted that we intentionally include studies of patient cohorts prior to vaccine availability, to avoid presenting unrepresentative comparisons of vaccinated and unvaccinated cohorts.

R1.5. Which sampling methodology was used. The statistical techniques used should be explained in one line and why did you use that technique particularly.

  • Authors: We clarify in section 2.2 (Eligibility Criteria) that all records meeting eligibility criteria were retained and that a sampling methodology was not used. Details on statistical techniques and justification for their use have been clarified in section 2.4 (Statistical Analyses).

R1.6. Figures are not adjusted. For example, it is difficult to see the figure 1. Moreover, there are two figures with label figure 1. Kindly change the figure numbers and adjust it in the explanation.

  • Authors: Thank you. This has been corrected.

R1.7. Conclusion section should also be extended with future perspectives and limitations of the study.

  • Authors: The conclusion has been expanded to include the strengths of the study, limitations, implications for healthcare practice, and future perspectives.
  1. Grammatical and sentence structure of overall study should be checked for the mistakes
  • Authors: Thank you. We have reviewed in full for grammatical and sentence structure errors.

R1.9. Overall study is acceptable for publication but after considering the above-mentioned changes.

  • Authors: Thank you again for taking the time to review. We have integrated your proposed changes and feel the manuscript has benefited from your feedback.

Reviewer 2 Report

In this study retrospective, the authors described the demographics, baseline clinical characteristics and treatments, and clinical outcomes during hospitalization among U.S. patients admitted to the hospital with at least one positive SARS-CoV-2 nucleic acid amplification test (NAAT) in the pre-vaccine phase of March to November 2020.

Major comments

There are three databases with very interesting information, but unfortunately, in the way the data is presented, there is no major contribution to what is known about COVID-19.

I suggest authors make comparisons (Tables 1, 2 and 3) between the 3 groups using crude OR and adjusted OR, or HRs with 95% CI, or risk differences.

A more exhaustive analysis will provide stronger conclusions for the centers where information was obtained.

Minor comments

Figure 2 is incomplete.

Author Response

Reviewer #2

In this study retrospective, the authors described the demographics, baseline clinical characteristics and treatments, and clinical outcomes during hospitalization among U.S. patients admitted to the hospital with at least one positive SARS-CoV-2 nucleic acid amplification test (NAAT) in the pre-vaccine phase of March to November 2020.

  • Authors: We would like to thank the reviewer for taking the time to review and provide feedback on the manuscript. We greatly appreciate their comments. We have outlined below the actions taken in response to each individual comment; these are numbered as “R2.x” (for Reviewer 2 and the comment number).

Major comments

R2.1. There are three databases with very interesting information, but unfortunately, in the way the data is presented, there is no major contribution to what is known about COVID-19.

  • Authors: Our hope is that, by presenting analyses from a larger patient cohort across multiple databases, we have increased the geographic representativeness of the patient cohort and generalizability of our findings relative to earlier studies which are often specific to small patient cohorts from a single tertiary center. As such, we feel that the data offer valuable insight into the clinical course of infection, patient characteristics and comorbidities, and hospital-resource utilization associated with severe COVID-19 infection over time. We have expended the Introduction and Conclusion sections to clarify this point.

R2.2. I suggest authors make comparisons (Tables 1, 2 and 3) between the 3 groups using crude OR and adjusted OR, or HRs with 95% CI, or risk differences.

  • Authors: There were concerns about the inherent quality of COVID-19 data in the early phases of the pandemic. Due to the observational nature of the study, we did not feel confident that we would be able to establish negative infection status with a sufficient degree of certainty. For this reason, all comparisons were meant to be descriptive, and no calculations of statistical significance were conducted.

R2.3. A more exhaustive analysis will provide stronger conclusions for the centers where information was obtained.

  • Authors: Please see response to R2.2. Further, our goal was to prioritize discussion of key patterns and findings, given the large amount of data generated. Additional descriptive comparisons are included in the Supplementary Material.

Minor comments

R2.4. Figure 2 is incomplete.

  • Authors: Thank you. We have reviewed to confirm that Figure 2 is complete (corrected Figure caption and sizing)

Reviewer 3 Report

In this manuscript authors described the demographics, baseline clinical characteristics and treatments, and clinical outcomes during hospitalization among U.S. patients admitted to the hospital with at least one positive SARS-CoV-2 nucleic acid amplification test (NAAT) in the pre-vaccine phase of March to November 2020.

The argument of the manuscript is original, but the manuscript has several methodological limitations. Inclusion and exclusion criteria must be clearly specified. Furthermore, discussion section lacks contents on the implications this study could have on national policies and what the paper adds to scientific literature. It is necessary that the authors investigate these aspects in depth, highlighting the areas in which these results could have healthcare implications. 

Especially results and discussion need a relevant revision both in stylistic terms and in contents. In general, the English in the paper can be understood.

The Major Essential Revisions include: 

-       In abstract and material and methods section it not clear why the study period initiated on February, while in the title and in the rest of the manuscript the month of March is referred to as the starting date of the study

-       In the materials and methods section authors should specify who carried out the data collection and in which place, describing these aspects in detail.

-       The exclusion criteria have been indicated in the materials and methods, but the inclusion criteria should also be made explicit. I suggest putting the inclusion and exclusion criteria in a separate paragraph and not in the statistical analyses section.

-       Line 120: authors should specify the meaning of the values in brackets. Do the values indicate the interquartile range? 75 and 95 percentiles?

-       I suggest to report two decimal values in the percentages both in the text and in the tables.

-       Line 123: I suggest to report the Software version used

-       Line 249-250: It would be important that the authors expand on this concept. In particular with regard to the comorbidities most frequently reported. This is a very interesting aspect due to the limitated scientific literature. For this reason I suggest to deepen this topic by citing more scientific articles. For example: Ioannou P, Spentzouri D, Konidaki M, Papapanagiotou M, Tzalis S, Akoumianakis I, Filippatos TD, Panagiotakis S, Kofteridis DP. COVID-19 in Older Individuals Requiring Hospitalization. Infect Dis Rep. 2022 Sep 12;14(5):686-693. doi: 10.3390/idr14050074. PMID: 36136824; PMCID: PMC9498435; Piazza MF, Amicizia D, Marchini F, Astengo M, Grammatico F, Battaglini A, Sticchi C, Paganino C, Lavieri R, Andreoli GB, Orsi A, Icardi G, Ansaldi F. Who Is at Higher Risk of SARS-CoV-2 Reinfection? Results from a Northern Region of Italy. Vaccines (Basel). 2022 Nov 8;10(11):1885. doi: 10.3390/vaccines10111885. PMID: 36366393; PMCID: PMC9692964.

-       Lines 293-298: It is important that the authors expand on the concept that this study and others can provide insights into real care needs and how professionals can support them to better cope in the post-Covid-19 phase, thus improving quality of life. For this reason I suggest to deepen this topic by citing more scientific articles. For example: Carola V, Vincenzo C, Morale C, Pelli M, Rocco M, Nicolais G. Psychological health in COVID-19 patients after discharge from an intensive care unit. Front Public Health. 2022 Aug 12;10:951136. doi: 10.3389/fpubh.2022.951136. PMID: 36033791; PMCID: PMC9411785; Piras I, Piazza MF, Piccolo C, Azara A, Piana A, Finco G, Galletta M. Experiences, Emotions, and Health Consequences among COVID-19 Survivors after Intensive Care Unit Hospitalization. Int J Environ Res Public Health. 2022 May 21;19(10):6263. doi: 10.3390/ijerph19106263. PMID: 35627801; PMCID: PMC9141708)

-       Authors should expand more the conclusion section especially explaining what this study adds to the scientific literature. It is necessary that the authors investigate these aspects in depth, highlighting the areas in which these results could have healthcare implications.

Author Response

Reviewer #3

In this manuscript authors described the demographics, baseline clinical characteristics and treatments, and clinical outcomes during hospitalization among U.S. patients admitted to the hospital with at least one positive SARS-CoV-2 nucleic acid amplification test (NAAT) in the pre-vaccine phase of March to November 2020.

The argument of the manuscript is original, but the manuscript has several methodological limitations. Inclusion and exclusion criteria must be clearly specified. Furthermore, discussion section lacks contents on the implications this study could have on national policies and what the paper adds to scientific literature. It is necessary that the authors investigate these aspects in depth, highlighting the areas in which these results could have healthcare implications. 

Especially results and discussion need a relevant revision both in stylistic terms and in contents. In general, the English in the paper can be understood.

  • Authors: We would like to thank the reviewer for taking the time to review and provide feedback on the manuscript. We greatly appreciate their comments and believe that the manuscript has improved as a result of their contribution. We have outlined below the actions taken in response to each individual comment; these are numbered as “R3.x” (for Reviewer 3 and the comment number).

R3.1. In abstract and material and methods section it is not clear why the study period initiated on February, while in the title and in the rest of the manuscript the month of March is referred to as the starting date of the study

  • Authors: The study period has been removed from the title in response to a comment from another reviewer. In Section 2.1, we clarify that the February 5 start period was chosen to align with the date that the FDA authorized NAAT for COVID-19 diagnosis. In Section 3.1, we clarify that no hospitalizations were identified until March 2020, which is why the Results deal with a March–November timeframe. We took the position that the March timeframe should be reported in the Results since it is a matter of the data available for analysis, not part of the study design.

R3.2. In the materials and methods section authors should specify who carried out the data collection and in which place, describing these aspects in detail.

  • Authors: A description of the three study databases has been provided in Section 2.1 (Study Population). Information on data collection has been added in Section 2.4 (Statistical Analyses). Author contributions have been provided in the Author Contributions section.

R3.3. The exclusion criteria have been indicated in the materials and methods, but the inclusion criteria should also be made explicit. I suggest putting the inclusion and exclusion criteria in a separate paragraph and not in the statistical analyses section.

  • Authors: As recommended, inclusion and exclusion criteria have been clarified in a new subsection (2.2 Eligibility Criteria).

R3.4. Line 120: authors should specify the meaning of the values in brackets. Do the values indicate the interquartile range? 75 and 95 percentiles?

  • Authors: We have revised to clarify that the values in brackets correspond to range of either 5th–95th percentiles or 25th–75thpercentiles, as specified in the Results section.

R3.5. I suggest reporting two decimal values in the percentages both in the text and in the tables.

  • We would suggest presenting one decimal value for percentages, since this is likely to be easier for readers to follow without sacrificing key insights from the data.

R3.6. Line 123: I suggest reporting the Software version used

  • Authors: We have added software versions as suggested.

R3.7. Line 249-250: It would be important that the authors expand on this concept. In particular with regard to the comorbidities most frequently reported. This is a very interesting aspect due to the limited scientific literature. For this reason I suggest to deepen this topic by citing more scientific articles. For example: Ioannou P, Spentzouri D, Konidaki M, Papapanagiotou M, Tzalis S, Akoumianakis I, Filippatos TD, Panagiotakis S, Kofteridis DP. COVID-19 in Older Individuals Requiring Hospitalization. Infect Dis Rep. 2022 Sep 12;14(5):686-693. doi: 10.3390/idr14050074. PMID: 36136824; PMCID: PMC9498435; Piazza MF, Amicizia D, Marchini F, Astengo M, Grammatico F, Battaglini A, Sticchi C, Paganino C, Lavieri R, Andreoli GB, Orsi A, Icardi G, Ansaldi F. Who Is at Higher Risk of SARS-CoV-2 Reinfection? Results from a Northern Region of Italy. Vaccines (Basel). 2022 Nov 8;10(11):1885. doi: 10.3390/vaccines10111885. PMID: 36366393; PMCID: PMC9692964.

  • Authors: This section has been expanded as suggested, drawing on the citation from Ioannou to highlight the risk of multiple overlapping comorbidities. We did not add further detail on the risks associated with laboratory values, as this was not part of our analysis (similarly, a Greek population may not be comparable to a U.S. patient cohort). We did not add the Piazza citation referenced, out of concern that the risk factors for reinfection presented by Piazza and colleagues could differ from those for hospitalization.

R3.8. Lines 293-298: It is important that the authors expand on the concept that this study and others can provide insights into real care needs and how professionals can support them to better cope in the post-Covid-19 phase, thus improving quality of life. For this reason I suggest to deepen this topic by citing more scientific articles. For example: Carola V, Vincenzo C, Morale C, Pelli M, Rocco M, Nicolais G. Psychological health in COVID-19 patients after discharge from an intensive care unit. Front Public Health. 2022 Aug 12;10:951136. doi: 10.3389/fpubh.2022.951136. PMID: 36033791; PMCID: PMC9411785; Piras I, Piazza MF, Piccolo C, Azara A, Piana A, Finco G, Galletta M. Experiences, Emotions, and Health Consequences among COVID-19 Survivors after Intensive Care Unit Hospitalization. Int J Environ Res Public Health. 2022 May 21;19(10):6263. doi: 10.3390/ijerph19106263. PMID: 35627801; PMCID: PMC9141708)

  • Authors: This has been added to the discussion of the healthcare implications of this study (in the conclusion section)

R3.9. Authors should expand more the conclusion section especially explaining what this study adds to the scientific literature. It is necessary that the authors investigate these aspects in depth, highlighting the areas in which these results could have healthcare implications.

  • Authors: The Conclusion has been expanded to include the study strengths (how it adds to the scientific literature), limitations, and implications for healthcare practice

Round 2

Reviewer 2 Report

In the way the authors present the results, they only show summative data. There is no further contribution. I suggest that analyzes be carried out to improve the conclusions of their results.

I suggest authors make comparisons (Tables 1, 2 and 3) between the 3 groups using crude OR and adjusted OR, or HRs with 95% CI, or risk differences.

A more exhaustive analysis will provide stronger conclusions for the centers where information was obtained.

Author Response

We thank the reviewer for raising this concern. From our careful assessment of the reviewer’s three comments, we felt that each addressed the same core concern and recommendation for inferential analyses, so we have chosen to provide a single response to the three comments.

While we have given careful consideration of the reviewer’s suggestions, we have decided it would not be appropriate to conduct the inferential analysis suggested across the reviewer's comments given the scope of our study. Briefly, the aim of our study is to characterize the clinical course of severe COVID-19 infection (i.e., requiring hospitalization) in the early, pre-vaccination phase of the pandemic. As such, we worry that inferential analyses comparing differences across the three databases could divert focus on the results provided and lead to confusion for the reader.

In addition, heterogeneity across databases would make findings from such analysis difficult to interpret. For example, each EHR data source has unique regional variations and healthcare system, and the population sample represented by each data source may be different. Thus, any significant differences detected because of underlying database differences may skew the results making them difficult to interpret.

Furthermore, to obtain the suggested adjusted risk estimates and make comparisons across databases, we would need to pool patient-level data across the data sources into one database. This is not possible due to patient privacy concerns and the data use agreements we have with some data partners do not allow direct access to these data.

For these reasons, we chose a priori to use a common standardized protocol to conduct data analyses at each individual data source and have described the demographics, baseline clinical characteristics and treatments, and clinical outcomes among hospitalized COVID-19-positive patients, confirmed by SARS-CoV-2 NAAT from each. We recognize the lack of adjustment for potential confounders as a limitation of our study and have added the following statement on page 11 of the manuscript ‘The presented findings are descriptive and as such do not take into account or adjust for potential confounding factors.’ Nevertheless, we believe that our findings provide valuable insight into the clinical course of infection, patient characteristics and comorbidities, and hospital-resource utilization associated with severe COVID-19 infection over time.

Reviewer 3 Report

Authors have addressed all suggestions requested. Thus I consider the manuscript ready for pubblication at this phase

Author Response

Thank you to the reviewer for taking the time to provide feedback on our manuscript. It has benefited from their comments.